# Grouping Methods of Cluster Dynamics Model for Precipitation Kinetics

**Kun Xu** [1] , **Brian G. Thomas** [2] , **Yueyue Wu** [1], **Haichuan Wang** [1], **Hui Kong** [3] **and Zhaoyang Wu** [1,*]

[1]  Key Laboratory of Metallurgical Emission Reduction & Resources Recycling (Anhui University of Technology), Ministry of Education, Ma'anshan 243002, China; kunxu@ahut.edu.cn (K.X.); ahutyueyuewu@163.com (Y.W.); which@ahut.edu.cn (H.W.)

[2]  Department of Mechanical Engineering, Colorado School of Mines, Golden, CO 80401, USA; bgthomas@mines.edu

[3]  International Science & Technology Cooperation Base for Intelligent Equipment Manufacturing under Special Work Environment, Anhui University of Technology, Ma'anshan 243002, China; konghui@ahut.edu.cn

*  Correspondence: ahutwzy@ahut.edu.cn; Tel.: +86-0555-231-1571

**Abstract:** Due to its simplicity and efficiency, cluster dynamics modeling has been widely used to simulate microstructure evolution in materials, such as defect formation in metals. However, its computation cost becomes prohibitive when the clusters grow too large, so a particle-size-grouping method is often required. In this paper, three different size-grouping methods are compared with the exact solution of the ungrouped cluster dynamics model for $Al_3Sc$ precipitation in an Al-0.18 at.% Sc alloy. A new assumption of logarithmically-linear distribution of cluster number densities inside each size group is shown to be the most efficient way to match with all results of the ungrouped model. Finally, the calculated results are compared with the measured sizes and distributions of $Al_3Sc$ precipitates at different aging temperatures. The new size-grouping method is shown to have better accuracy for the chosen discretization and time-stepping method evaluated. This will enable significant computational savings, and the extension of time scales and cluster sizes to the ranges of realistic metallurgical systems, while preserving reasonable accuracy.

**Keywords:** precipitation kinetics; cluster dynamics model; particle-size-grouping method; Al-Sc alloy

---

## 1. Introduction

Cluster dynamics (CD) is a powerful tool to model microstructure evolution in materials, such as the nucleation, growth and coarsening of precipitates [1–12], point defects [13–20], bubbles [21–23], or inclusions [24–26]. Compared with other models, the CD model has several unique advantages for describing precipitation kinetics, especially in irradiated materials or at an early stage of precipitation. First, it describes the entire evolution of every cluster as a single continuous and competing process, using the same set of equations and physically-based properties, so the different phenomena of incubation, nucleation, growth, and coarsening all arise naturally. In addition, all particle sizes, ranging from single atoms/molecules, unstable embryos, and stable nuclei, to coarsening particles, can be tracked simultaneously. Therefore, the calculated results are easy to compare with experimental measurements of various resolution limits, even when clusters smaller than the stable nucleus can be detected. Finally, no extra explicit empirical laws or fitting parameters are required. For example, Becker–Döring nucleation theory [27,28], which is very important for many kinetic models, such as modified Langer-Schwartz [29,30], Kampmann-Wagner Numerical [31,32], or Matcalc [33,34], is not needed in the CD model.

However, the original CD model is based on simple incrementing of the number of atoms or molecules in a cluster, which always encounters computational difficulties when the simulated cluster size becomes too large [22–25]. A particle-size-grouping (PSG) method is thus often introduced to save computational cost without sacrificing too much accuracy. In this method, the clusters are sorted into dozens or hundreds of size groups, and each group covers a characteristic cluster size range [13–17,23–26]. Instead of tracking every cluster size, with its unique individual number of molecules, the number of clusters inside each group, defined by its size range, is then evaluated. This requires an assumption regarding the distribution of cluster number densities inside each size group. In previous works, a uniform [13–15,23–25] or linear [16,17] distribution was frequently assumed, but none of these approaches have been rigorously shown to produce accurate results. Although many grouping methods have been developed to enable practical simulations of large-scale precipitation, the calculated results have mainly been compared directly with experiments, and rarely have been verified with the exact solution of the original ungrouped CD model. Therefore, the accuracy of these grouping methods has not been properly evaluated until the present work.

The purpose of this paper is to develop and demonstrate a more accurate and efficient PSG CD method. First, three grouping methods are introduced, which assume uniform, linear, and log-linear distributions of cluster number densities inside each size group. Both ungrouped and grouped CD models are applied to simulate the precipitation of $Al_3Sc$ in an Al-0.18 at.% Sc alloy, and the accuracies of these methods are carefully evaluated by comparing their predictions with the exact solution of the ungrouped model. Finally, the evolutions of the mean precipitate sizes and precipitate size distributions are compared with experimental measurements.

## 2. Cluster Dynamics Model

In the CD model, clusters are assumed to be spherical-shaped precipitates of a secondary-phase material, and the total volume of all clusters is assumed to be negligible compared with the matrix material. For a precipitation process in the solid matrix, the growth and shrinkage of all clusters are assumed to be adiabatic and caused only by the movement of single atoms or molecules. For size $i$ clusters, which contain $i$ molecules (which could be atoms) of the secondary phase, the evolution of their number per unit volume, defined as "number density", $n_i$, satisfies the rate equations:

$$\frac{\partial n_i}{\partial t} = J_{i-1} - J_i, \quad i \geq 2,$$ (1)

Here $t$ is the time, $J_i$ is the mass flux from size $i$ clusters to size $i + 1$ clusters, which is defined as

$$J_i = n_1 \beta_i n_i - \alpha_{i+1} n_{i+1},$$ (2)

Here $\beta_i$ and $\alpha_i$ are the growth rate of molecules joining a size $i$ cluster, and the dissolution rate of those leaving. These equations indicate how size $i$ clusters decrease in number due to their own growth after capturing one molecule, or due to their own shrinkage after losing one molecule. They may also be generated by the growth of size $i - 1$ clusters after capturing one molecule, or by the shrinkage of size $i + 1$ clusters after losing one molecule. Capturing or losing more than a single molecule at a time is called "collision" [24–26], or "breakup", and is not considered in the diffusion-controlled kinetic behavior of this work.

The evolution of number density of single molecules is given as

$$\frac{\partial n_1}{\partial t} = -\sum_{i=1}^{i_M} (1 + \delta_{1,i}) J_i = -2J_1 - \sum_{i=2}^{i_M} J_i,$$ (3)

where $\delta_{i,j}$ is the Kronecker delta, $\delta_{i,j} = 1$ when $i = j$, and $\delta_{i,j} = 0$ when $i \neq j$, and $i_M$ is the number of molecules contained in the largest cluster. This equation states that single molecules are always

consumed by cluster growth, and generated by cluster shrinkage. If $i_M$ is chosen to be large enough and number density, $n_{i_M}$, is infinitely small, the total mass of precipitate phase is proven to be conserved because

$$\sum_{i=1}^{i_M} i \frac{dn_i(t)}{\partial t} = -(i_M + 1) J_{i_M} = -(i_M + 1) \beta_{i_M} n_1 n_{i_M} \approx 0, \tag{4}$$

To accurately model precipitation behavior, both rates in Equation (2) need to be well determined. The growth rate, also called the "condensation rate" [5,6,8–10], "impingement rate" [1], "capture rate" [2,3,23], or "absorption rate" [4,11,12,19–21], is generally defined from the concentration gradient near a size $i$ cluster surface, as follows [1,5,6,8–10,21,25,26,35]

$$\beta_i = 4\pi D r_i, \tag{5}$$

where $r_i$ is the radius of the size $i$ cluster, and $D$ is the diffusivity of the secondary (precipitating) phase within the matrix phase. The growth rate is shown to increase linearly with the cluster size. If the precipitate phase contains more than one alloying element, the precipitation rate is controlled by the diffusion coefficient of the atoms of the slowest-diffusing element.

The dissolution rate, also called the "emission rate" [1–4,8,10–12,21] or "evaporation rate" [5,6,9,23], is generally determined using one of two methods. The first method is based on the Gibbs–Thomson equation, which states that the equilibrium concentration near a cluster surface is a function of the cluster size [36], as follows

$$n_{1,i}^{eq} = n_1^{eq} \exp\left(\frac{2\gamma V_m^P}{R_g T} \frac{1}{r_i}\right), \tag{6}$$

where $n_{1,i}^{eq}$ and $n_1^{eq}$ are the equilibrium concentrations of completely dissolved single molecules of precipitate phase at the surface of a size $i$ cluster and with a planar interface, respectively, $\gamma$ is the interface free energy between the precipitate and matrix phases, $V_m^P$ is the molar volume of precipitate phase, $R_g$ is the gas constant, and $T$ is the absolute temperature. Two equilibrium concentrations satisfy $n_{1,i\to\infty}^{eq} = n_1^{eq}$ when cluster size is infinitely large. At equilibrium, the growth velocity of the size $i$ cluster is assumed to be zero, so the dissolution rate is thus given as [25,26,35]

$$\alpha_i = \beta_i n_{1,i}^{eq} = \beta_i n_1^{eq} \exp\left(\frac{2\gamma V_m^P}{R_g T} \frac{1}{r_i}\right), \tag{7}$$

The other method is based on Gibbs free energy. The total free energy of a cluster consists of a volume-related part and an interface-related part. When a size $i$ cluster dissolves into a size $i - 1$ cluster and a single molecule, its volume-related energy stays unchanged, but the interface-related energy increases. The dissolution rate is thus expressed as [1,3–6,8]

$$\alpha_i = \beta_{i-1} n_1^{eq} \exp\left\{\frac{4\pi \gamma_i r_i^2 - 4\pi \gamma_{i-1} r_{i-1}^2}{kT}\right\}, \tag{8}$$

where $k$ is Boltzmann's constant, and $\gamma_i$ is the interface free energy between a size $i$ cluster and matrix phases. To find the expression in Equation (8), $n_1^{eq}$ needs to be expressed as a function of the interface free energy, such as that in Clouet's work [6]. However, the value from that equation is never adopted into the practical computation, and $n_1^{eq}$ is actually obtained from a thermodynamic equilibrium analysis instead. Alternatively, the same expression can be newly explained and obtained in another, more reasonable way, as follows [4].

If the crystal structure of precipitate phase is face-centered cubic with a lattice parameter of $a$, molar volume $V_m^P = N_A a^3 / 4$, cluster size $r_i = (3V_i/4\pi)^{1/3} = \left(3/4\pi \cdot i a^3/4\right)^{1/3} = (3i/16\pi)^{1/3} a$.

Furthermore, if the interface free energy is assumed independent of the cluster size, both Equations (7) and (8) lead to the following expression:

$$\alpha_i/\beta_i \approx n_1^{eq} \exp\left\{\left(\frac{2\pi}{3}\right)^{1/3}\frac{\gamma a^2}{kTi^{1/3}}\right\} \ for \ i \gg 1, \tag{9}$$

A similar statement holds for body-centered cubic or hexagonal close-packed structures, except for using a coefficient of $(8\pi/3)^{1/3}$ or $(3\pi)^{1/3}$ instead of $(2\pi/3)^{1/3}$ in Equation (9). This demonstrates that the two dissolution rates from Equations (7) and (8) are often identical, especially for very large clusters.

Equations (1)–(3), (5), and (7) or (8) comprise the exact solution of the original ungrouped CD model.

## 3. Size Grouping Methods

The system of differential rate equations that comprise the ungrouped CD model presented in Section 2 can be integrated directly by a suitable numerical method to obtain the complete evolution map of all clusters. However, the computation becomes infeasible when the clusters eventually grow too large, which often happens in real material systems. Since the ungrouped CD model in Equations (1)–(3) is based on a simple increment of molecule number, it will inevitably experience difficulties in computational storage and execution time for realistic large-scale simulations. To overcome these difficulties, a size grouping method is often introduced.

The main objective of the size-grouping method is to reduce the number of the unsolved variables and population-balance rate equations by collecting the clusters into size groups, each containing a range of cluster sizes. Instead of tracking every individual cluster size, the clusters are tracked using the much smaller number of size groups. Increasing the range of cluster sizes in each size group exponentially has the benefit of enabling a simulation to span the complete realistic size range, from single molecules to realistic precipitates containing millions of molecules, with only several hundred size groups. To achieve this benefit, however, the mass transfer inside each size group and between the reacting size groups needs to be precisely quantified, in order to satisfy the mass balance and the related kinetic rules.

Generally, each size group has the number range $\Delta x_j = x_j - x_{j-1}$, where the subscript $j$ indicates the number of different cluster sizes in group $j$, which includes all cluster sizes containing any number of molecules ranging from $x_{j-1} + 1$ to $x_j$. The total number density of clusters accumulated in the $j - th$ size group, $N_j$, is thus determined as

$$N_j = \sum_{i=x_{j-1}+1}^{x_j} n_i = \sum_{k=1}^{\Delta x_j} n_{(x_{j-1}+k)}, \tag{10}$$

The arithmetic mean number of molecules in the $j - th$ size group of clusters is

$$\langle x \rangle_j = \frac{1}{\Delta x_j}\sum_{i=x_{j-1}+1}^{x_j} i = \frac{1}{\Delta x_j}\sum_{k=1}^{\Delta x_j}(x_{j-1}+k) = \frac{x_{j-1}+1+x_j}{2}, \tag{11}$$

Note that the limiting case of $\Delta x_j = 1$, gives $\langle x \rangle_j = x_j = j$. Thus, the ungrouped CD model is just a limiting case of this grouping method.

A method to define the size group number ranges is often suggested as

$$x_j = x_{j-1} + ceil\left[\lambda(x_{j-1}+1)\right], \tag{12}$$

where $\lambda$ is a fractional number, and function ceil($x$) gives the smallest integer larger than the real number $x$. Slightly revised from a previous work [17], this grouping method starts from $x_1 = 1$,

and guarantees that $x_j$ is always an integer and never smaller than its group index $j$. By choosing a small value of $\lambda$, $\Delta x_j = 1$ holds for small size groups, and then gradually and exponentially increases with the group index. The entire cluster size spectrum is thus divided into two continuous regimes. The first regime starts from single molecules, and simply increments the number of molecules in each size group. The second regime immediately follows the discrete regime, and consists of size groups which each contain clusters of more than one size. The two regimes are distinguished by whether $\Delta x_j = 1$ or greater. Every cluster size is included in exactly one size group.

In any grouping method, the details that govern behavior inside each size group need to be defined prior to the calculation. Three different assumed distributions of cluster number densities within each size group are suggested, as follows:

(1) Uniform distribution

The simplest assumed distribution of cluster number densities inside each size group is, obviously, uniform. This method was first suggested by Kiritani [13], and later investigated by Korwa [14], Hayns [15], and Golubov et al. [16] for modeling the kinetics of point-defect clusters. For a large size group range, $\Delta x_j \gg 1$, the original discrete CD model can be written in a continuous scheme by Taylor expansion, as follows

$$\frac{\partial n(x,t)}{\partial t} \begin{aligned} &= J(x-1,t) - J(x,t) = \beta_{x-1}n_1 n_{x-1} - \alpha_x n_x - \beta_x n_1 n_x + \alpha_{x+1}n_{x+1} \\ &\approx \frac{\partial}{\partial x}\left[(-\beta n n_1 + \alpha n) + \frac{\partial}{\partial x}(\beta n n_1 + \alpha n)\right], \ x \geq 2 \end{aligned} \tag{13}$$

This is the form of the well-known Fokker–Planck equation, which has been widely studied in the kinetic simulation of particles or defects [2,18–20,22]. The above equation is discretized again in the size-group scale as

$$\frac{\partial N_j}{\partial t} = \frac{2\beta_{j-1}N_1 N_{j-1}}{\Delta x_{j-1} + \Delta x_j} - \frac{2\alpha_j N_j}{\Delta x_{j-1} + \Delta x_j} - \frac{2\beta_j N_1 N_j}{\Delta x_j + \Delta x_{j+1}} + \frac{2\alpha_{j+1}N_{j+1}}{\Delta x_j + \Delta x_{j+1}}, \quad j \geq 2, \tag{14}$$

where the total number density of clusters, $N_j = \bar{n}_j \Delta x_j$, and $\bar{n}_j$ is the average cluster number density inside the $j-$th size group. Comparing with ungrouped Equations (1) and (2), a noticeable change is using a space step of $(\Delta x_j + \Delta x_{j+1})/2$ for cluster growth and shrinkage between group $j$ and $j + 1$. To satisfy the mass balance of both clusters and molecules, the population equation of single molecules is special, and is given as

$$\frac{\partial N_1}{\partial t} = \frac{\partial n_1}{\partial t} = -\sum_{k=1}^{\infty} \frac{2}{\Delta x_k + \Delta x_{k+1}}\left(\langle x \rangle_{k+1} - (1 - \delta_{1,k})\langle x \rangle_k\right)(N_1 \beta_k N_k - \alpha_{k+1}N_{k+1}), \tag{15}$$

The growth and dissolution rates are taken from their values exactly at the group center as

$$\beta_j = \beta_{\langle x \rangle_j}, \ \alpha_j = \alpha_{\langle x \rangle_j}, \tag{16}$$

Equations (14)–(16) comprise the Kiritani PSG method, with a uniform distribution assumed inside each size group.

(2) Linear distribution

The uniform distribution is usually too simple to match reality. To improve accuracy, a linear distribution of cluster number densities within groups was proposed by Golubov and Ovcharenko [16,17], and later by Golubov et al., to model two dimensional helium-vacancy cluster evolution [23]. In these works, the cluster number densities inside the $j-$th size group were assumed as [16,17]:

$$n_i = L_0^j + L_1^j(i - \langle x \rangle_j), \ x_j \geq i \geq x_{j-1} + 1, \tag{17}$$

Summing the above distribution from $i = x_{j-1} + 1$ to $x_j$ inside the $j -$ th size group, the total number density of clusters in the $j -$ th size group evolves as

$$\frac{\partial N_j}{\partial t} = \sum_{i=x_{j-1}+1}^{x_j} \frac{\partial n_i}{\partial t} = \sum_{i=x_{j-1}+1}^{\Delta x_j} \frac{\partial \left[ L_0^j + L_1^j (i - \langle x \rangle_j) \right]}{\partial t} = \frac{\partial L_0^j}{\partial t} \Delta x_j = J_{x_{j-1}} - J_{x_j}, \tag{18}$$

This indicates that the evolution of cluster number density of the $j -$ th group is determined only by the mass fluxes near its two group boundaries. Multiplying Equations (1) and (14) by $i$, the total number density of molecules accumulated in the $j -$ th group, $S_j$, evolves as follows:

$$\frac{\partial S_j}{\partial t} = \sum_{i=x_{j-1}+1}^{x_j} i \frac{\partial n_i}{\partial t} = \sum_{i=x_{j-1}+1}^{x_j} i \frac{\partial \left[ L_0^j + L_1^j(i - \langle x \rangle_j) \right]}{\partial t} = \frac{d \left[ L_0^j \langle x \rangle_j + L_1^j \sigma_j^2 \right]}{dt} \Delta x_j = (x_{j-1} + 1) J_{x_{j-1}} - x_j J_{x_j} + \sum_{i=x_{j-1}+1}^{x_j-1} J_i \tag{19}$$

$$where \ \sigma_j^2 = \langle x^2 \rangle_j - \langle x \rangle_j^2 = \frac{1}{\Delta x_j} \sum_{i=x_{j-1}+1}^{x_j} i^2 - \frac{1}{\Delta x_j^2} \left( \sum_{i=x_{j-1}+1}^{x_j} i \right)^2, \tag{20}$$

Equating $N_j$ and $S_j$ in both ungrouped and grouped methods, the evolution equations for the distribution parameters, $L_0^j$ and $L_1^j$, are given as

$$\frac{\partial L_0^j}{\partial t} = \frac{J_{x_{j-1}} - J_{x_j}}{\Delta x_j}, \tag{21}$$

$$\frac{\partial L_1^j}{\partial t} = -\frac{(\Delta x_j - 1)}{2\sigma_j^2 \Delta x_j} \left[ J_{x_{j-1}} + J_{x_j} - \frac{2}{(\Delta x_j - 1)} \sum_{i=x_{j-1}+1}^{x_j-1} J_i \right] = -\frac{(\Delta x_j - 1)}{2\sigma_j^2 \Delta x_j} \left[ J_{x_{j-1}} + J_{x_j} - 2J_j^* \right], \tag{22}$$

where $J_j^*$ is the mean mass flux, excluding the value on the right group border in the $j -$ th group, and is defined as

$$J_j^* = \frac{1}{\Delta x_j - 1} \sum_{i=x_{j-1}+1}^{x_j-1} J_i, \tag{23}$$

The number density of the special case j = 1 for single molecules evolves as follows, ($L_1^1 = 0$ since $\Delta x_1 = 1$)

$$\frac{\partial N_1}{\partial t} = \frac{\partial L_0^1}{\partial t} = -2J_1 - \sum_{j=2}^{G_M} \left[ J_{x_j} + (\Delta x_j - 1) J_j^* \right], \tag{24}$$

where $G_M$ is the number of the largest size group, which can be determined by both the largest number of molecules, $i_M$, and the specific grouping method. Note that minor errors in previous works [16,17] have been corrected here in Equation (22) (in [16], a multiplier $-1/\Delta x_j$ in the right-hand side was lost, and the signs before $J_{x_j}$ and $J_j^*$ were exactly opposite in [17]). Moreover, note that $L_1^j = 0$ and $J_j^* = 0$ when $\Delta x_j = 1$, so these equations readily simplify to the limiting case of a single cluster size in each size group (i.e., the exact ungrouped CD model).

Unfortunately, the calculation of $J_j^*$ with Equation (23) was not clearly given in any previously published literature study. Simply traversing every increment of molecule number inside the entire $j -$ th group in Equation (17) is obviously too computationally expensive for a realistic application. In the current work, a relatively simple and accurate estimation of $J_j^*$ is proposed by summing the mass fluxes of all clusters, except for those at group borders that will jump into the neighboring groups after

growth or shrinkage, based on the growth and dissolution rates at the center of the group size range, as follows

$$(\Delta x_j - 1)J_j^* = \sum_{k=1}^{\Delta x_j - 1} J_{x_{j-1}+k} \approx \beta_{\langle x \rangle_j - 1/2} n_1 \left( N_j - n_{x_j} \right) - \alpha_{\langle x \rangle_j + 1/2} \left( N_j - n_{x_{j-1}+1} \right), \tag{25}$$

For the linear distribution, the cluster number densities can either increase or decrease monotonically within each size group. This requires the size group number range to be small enough to approximate the actual distribution, if it is nonmonotonic. However, this information is unknown before the calculation. Most important of all, the number densities obtained by this linear assumption are not guaranteed to be always non-negative. Although it was claimed not to lead to any significant errors for the total properties of point-defect clusters [17,18], the negative number densities obviously violate physical laws.

Equations (17), and (21)–(25) comprise the G-O PSG method, with a linear distribution assumed inside each size group.

(3) Logarithmically-linear distribution

From the above analysis, an ideal grouping method should satisfy three basic physical requirements. First, a summation of the hypothetical cluster number density distribution over any size group should be equal to its total number density, as defined by Equation (10). The higher order of mass conservation, and total number density of molecules, should also be satisfied. Next, physics requires that every cluster number density from the assumed distribution should be non-negative. Finally, the molecule number density distribution near size group boundaries should be continuous across neighboring size groups. In practice, it is difficult to find a distribution which meets all of these requirements.

A simple way to guarantee the non-negative cluster number densities within every group is to assume a log-linear distribution inside each size group. Following a similar idea to previous work by the authors [37], cluster number densities inside the $j - $th group are assumed as

$$n_i = n_{x_{j-1}+1} q_j^{i - x_{j-1} - 1} = n_{\langle x \rangle_j} q_j^{i - \langle x \rangle_j}, \quad x_j \geq i \geq x_{j-1} + 1, \tag{26}$$

where $q_j$ is a common ratio of the geometric progression. Applying the above assumed distribution into Equation (10), the total number density of clusters in the $j - $th size group is given as

$$N_j = n_{x_{j-1}+1} \frac{1 - q_j^{\Delta x_j}}{1 - q_j}, \tag{27}$$

The corresponding total number density of molecules in the $j - $th size group is

$$S_j = \frac{n_{x_{j-1}+1}}{1 - q_j} \left[ x_{j-1} + 1 - x_j q_j^{\Delta x_j} + \frac{q_j - q_j^{\Delta x_j}}{1 - q_j} \right], \tag{28}$$

Removing $n_{x_{j-1}+1}$ from Equations (27) and (28), the equation to find $q_j$ is written as

$$\left| \frac{S_j}{x_j N_j} - \frac{\left(1 - q_j\right)\left(x_{j-1} + 1 - x_j q_j^{\Delta x_j}\right) + q_j - q_j^{\Delta x_j}}{x_j \left(1 - q_j\right)\left(1 - q_j^{\Delta x_j}\right)} \right| < \varepsilon_{tol}, \tag{29}$$

In this equation, limiting values are given as $S_j/N_j = x_{j-1} + 1$ when $q_j = 0$, and $S_j/N_j = x_j$ when $q_j \to +\infty$. A binary search is performed until the absolute error of the above equation is smaller than a given tolerable error. The calculation starts from the smallest size group, where $n_{x_{j-1}+1} = n_{x_j} = N_j$ for $\Delta x_j = 1$. When the size group number increases to make $\Delta x_j > 1$, (inside second regime), $q_j$ is solved

using Equation (29). According to the assumed distribution, the number density of clusters at the left border of the $j-$ th size group is then calculated as

$$n_{x_{j-1}+1} = \frac{N_j(1-q_j)}{1-q_j^{\Delta x_j}},\tag{30}$$

Here $J_j^*$ is still estimated by Equation (25). Equations (18), (19), (25), (26), (28) and (29) comprise the new PSG method, which features a log-linear distribution of cluster number densities assumed within each size group.

## 4. Practical Considerations

(1) Time step size

As shown in Equations (1) and (2), the number density evolution of size $i$ clusters depends on its own magnitude. For ease of comparison, this work adopts a simple explicit time-integration scheme for all CD methods evaluated. This requires a small time step size to avoid numerical instability due to $n_i$ becoming unphysically negative if the dissolution of size $i$ clusters is overestimated due to an excessive time step size. This problem can be overcome by using an unconditionally-stable implicit scheme or an adaptive time step in the explicit scheme, which can be implemented in any of these CD methods. At time step $k$ of an explicit scheme, with an initial step size $\Delta t^k$, the following values are evaluated

$$\xi^k = \min_{i_M \geq i \geq 1}\left(\xi_i^k\right) = \min_{i_M \geq i \geq 1}\left(\frac{n_i^k - n_i^{k-1}}{n_i^{k-1}}\right) for\ n_i^{k-1} \neq 0,\tag{31}$$

For $n_i^{k-1} = 0$, it is possible that $n_i^k < 0$, so $\xi_i^k = -1.1$ under this situation. For other cases of $n_i^{k-1} = 0$, $\xi_i^k = 0$ is stated. If $\min_{i_M \geq i \geq 1}\left(n_i^k\right) < 0$, then $\xi^k < -1$, which means that the number densities of certain clusters are negative, and a smaller time step is thus necessary to redo the integration at time step $k$. Since the change of cluster number density is linearly proportional to the time step size, a new step size is reasonably suggested as

$$\left(\Delta t^k\right)_{new} = -\frac{0.9}{\xi^k}\Delta t^k,\tag{32}$$

Here 0.9 is a safety factor to ensure positive cluster number densities with the new time step size. If $\xi^k < -1$ is obtained, all rate equations need to be reintegrated with this new step size.

Meanwhile, mass conversation requires that cluster number densities cannot all increase or all decrease simultaneously. If $\min_{i_M \geq i \geq 1}\left(n_i^k\right) \geq 0$, then $0 > \xi^k \geq -1$, and all number densities are nonnegative and the calculation can be safely moved to the next time step $k+1$. For computational efficiency, the next time step size is suggested as

$$\Delta t^{k+1} = \min\left(-\frac{0.9}{\xi^k}, 1.1\right)\Delta t^k,\tag{33}$$

The same time step size restrictions need to be applied to the PSG CD methods. Moreover, $x_j \geq S_j/N_j \geq x_{j-1} + 1$ must always be satisfied, to prevent problems within the limiting cases. If the time step size chosen is too large, these constraints are likely to be violated. Assuming $x_j \geq S_j^k/N_j^k \geq x_{j-1} + 1$ has been true at time step k, then $S_j^{k+1}$ and $N_j^{k+1}$ at the next time step $k+1$ must satisfy

$$x_j \geq \frac{S_j^{k+1}}{N_j^{k+1}} = \frac{S_j^k + \left[(x_{j-1}+1)J_{x_{j-1}} - x_j J_{x_j} + (\Delta x_j - 1)J_j^*\right]\Delta t_j^k}{N_j^k + \left[J_{x_{j-1}} - J_{x_j}\right]\Delta t_j^k} \geq x_{j-1} + 1,\tag{34}$$

This leads to the following constraints on the time step size

$$
\begin{cases}
\Delta t_j^k \leq \dfrac{S_j^k - (x_{j-1}+1)N_j^k}{(\Delta x_j - 1)(J_{x_j} - J_j^*)} & if \ J_{x_j} > J_j^* \\[4mm]
\Delta t_j^k \leq \dfrac{x_j N_j^k - S_j^k}{(\Delta x_j - 1)(J_j^* - J_{x_{j-1}})} & if \ J_j^* > J_{x_{j-1}}
\end{cases}, \tag{35}
$$

These calculations need to loop through all size groups, and the maximum time step is determined as $\Delta t^k = 0.9 \min\limits_{G_M \geq j \geq 1} \left(\Delta t_j^k\right)$, where 0.9 is a safety factor. Actually, the time step size determined by Equations (31)–(35) can ensure that all cluster number densities are non-negative, even for the assumed linear distribution of cluster number densities. However, a poor assumption for the distribution will make the calculated time step size become too small, and eventually tend towards zero, and the calculation will almost stop. Therefore, the smallest time step size is always set and applied in practical calculations. This makes the calculation run smoothly, but negative number densities cannot be completely avoided anymore.

(2) Initializing calculation of cluster number density

The number of molecules, $i_M$, or groups, $G_M$, composing the largest cluster is set as a boundary value with a constant zero number density all through the calculation. In order to guarantee the chosen $i_M$ or $G_M$ is large enough to conserve mass according to Equation (4), the number density of the second largest cluster, $i_{M-1}$, or group, $G_{M-1}$, needs to be sufficiently small. A safe judging criterion is suggested as

$$
n_{i_M-1} < 1 \#/m^3 \quad or \quad N_{G_M-1} < \Delta x_{G_M-1} \#/m^3, \tag{36}
$$

To save computational memory and time, the initial number of cluster sizes can be set as a small number, and the maximum cluster size group number increased by one, whenever Equation (36) is violated. In this work, the maximum size of molecules or groups is fixed to be constant through the entire calculation, but the evolution of a given cluster or group is calculated only when its cluster number density becomes larger than the minimum given in Equation (36). A reasonable criterion for calculating $n_i$ or $N_j$ at time step $k + 1$ is suggested by considering the value of its left neighbor size group at the previous time step $k$. If $n_{i-1}^k > 1/m^3$ or $N_{j-1}^k > \Delta x_{j-1}/m^3$ is satisfied, $n_i^{k+1}$ or $N_j^{k+1}$ will be calculated. Otherwise, its calculation is ignored.

If the numerical cluster number density in a size group is too small, this means that these clusters seldom exist. Furthermore, the calculation of these tiny values is time-consuming, and the time step size determined by Equations (31)–(35) for those rare clusters might become unnecessarily small. Therefore, after finishing the integration, if abs $(n_{i-1}^k) < 1/m^3$ or abs $(N_{j-1}^k) < \Delta x_j/m^3$, these tiny number densities are reset to zero again.

(3) Comparison with experimental measurement

The CD model tracks clusters of every size, including single molecules. However, for any given experimental technique, there exists a resolution limit, meaning that clusters smaller than this limit cannot be detected. In order to compare the numerical results with measurements, a minimum truncating cluster size, $r_{tr}$, (with the corresponding index of molecule $i_{tr}$ or size group $G_{tr}$) is chosen according to the resolution limit, and the total number density and mean radius of detectable precipitates (clusters) are defined as

$$
N_P = \sum_{i \geq i_{tr}}^{i_M-1} n_i \ (ungrouped) \quad or \quad N_P = \sum_{j \geq G_{tr}}^{G_M-1} N_j \ (grouped), \tag{37}
$$

$$
\langle r \rangle_P = \sum_{i \geq i_{tr}}^{i_M-1} r_i n_i / N_P \ (ungrouped) \quad or \quad \langle r \rangle_P = \sum_{j \geq G_{tr}}^{G_M-1} \langle r \rangle_j N_j / N_P \ (grouped), \tag{38}
$$

By these calculations, the results of one numerical simulation can be compared with various experimental measurements, even if they have different resolution limits, so long as the experimental conditions are otherwise the same.

## 5. Test Problem: Al-Sc Alloy Precipitation

CD models have limitations in many real applications. For example, the clusters are usually not uniformly distributed in the matrix phase. A kinetic Monte Carlo method may be capable of overcoming this problem for a small computational domain [5,8–11,38]. Most importantly, the CD models described above are derived assuming homogeneous precipitation. For most real alloys, however, the favorable sites for precipitation are usually dislocations, grain boundaries, secondary phase particles, or other crystal defects, so a heterogeneous precipitation model would be better.

The $Al_3Sc$ in Al-Sc alloy has been shown to precipitate out homogeneously as coherent, spherical particles, with a very small fraction, especially when the $Al_3Sc$ size is small. Thus, this system has been widely studied by many researchers in the last few decades [5–8,38–50]. In the laboratory experiments [39,40,42,44,48,49], pure Sc or high-Sc Al-Sc alloy was added to a pure Al melt, and stirred to achieve a good mixture, with the desired Sc composition under an inert atmosphere. The melt was then cast and cold rolled into very thin strips, with a thickness of several hundred microns. These specimens were homogenized at a solution temperature below the melting point for hours to produce a solid solution without $Al_3Sc$ precipitates, and then quenched directly to the precipitation temperature and held for various lengths of time. When the desired aging time was reached, the specimens were quickly quenched again to room temperature [39,42,44]. As precipitation occurs mainly at the aging temperature, and the measurements were made at room temperature, the relevant properties, such as solubility, diffusion coefficient, and interface free energy, etc., are taken at the aging temperature, for the modeling of this experiment.

The crystal structure of $Al_3Sc$ is $L1_2$, which is face-centered cubic with 8 Sc atoms at the cubic corners, and 6 Al atoms at the facial centers [38,44,45,48,50]. At room temperature, the lattice parameter of $Al_3Sc$ is a = 0.4105 nm, which is only 1.3% larger than that of the Al matrix, and the molar volume of $Al_3Sc$ is $V_m^P = 4.166 \times 10^{-5}$ m$^3$/mol [39,41,42,44,48,50]. The radius of a size $i$ cluster is $r_i = (3V_i/4\pi)^{1/3} = (3i/4\pi)^{1/3}a$. For the Al-rich side of the Al-Sc binary phase diagram, the equilibrium molar concentration of Sc in Al is $C_{Sc}^{eq} = \exp(2.25)\exp(-7766/T)$ for the temperature range of 643–803 K [40]. The diffusion of Sc in Al is taken as $D_{Sc}$ (m$^2$/s) $= 5.31 \times 10^{-4}\exp(-173{,}000/R_gT)$ from Fujikawa's work [47], which lies between data of Marquis [39] and Watanabe [42]. The best fit of interface free energy between the calculation and the experiment was found to be around $\gamma = 0.12$ J/m$^2$, which agrees well with values previously reported: $0.094 \pm 0.023$ J/m$^2$ for nucleation (561–616 K, Hyland [45]), 0.105 J/m$^2$ for coarsening (623 K, Novotny [44]), 0.12 J/m$^2$ for the coherent interface free energy (673–763 K, Iwamura [48]), and an average value of 0.108 J/m$^2$ for various cluster sizes when modeling precipitation (573–673 K, Clouet [6]).

The precipitation of $Al_3Sc$ in an Al-0.18 at.% Sc alloy aging at 300 °C was chosen for CD modeling to evaluate the accuracy of the three grouping methods described in Section 3. A fourth order Runge-Kutta integration scheme was used, and the maximum number of molecules in a cluster was chosen as $i_M = 16{,}000$, which corresponds to a maximum cluster radius of $r_{max} \approx 6.42$ nm and guarantees $n_{i_M-1} = 0$ at the terminating precipitation time t $= 1 \times 10^7$ s. According to Equation (12) with $\lambda = 0.05$, at least $G_M = 149$ size groups are necessary to cover this largest cluster size. The initial and minimum time step size in integration were both set as 1 s, which have been demonstrated to stabilize the ungrouped model with good accuracy. It was unnecessary to further decrease this value, since certain grouping methods themselves may not guarantee all non-negative cluster number densities for a smooth calculation, no matter how small the minimum time step size is.

The calculated emission rates from Equations (7) and (8) are shown in Figure 1a. There is no emission rate for single molecules, since they cannot dissolve any further. For very small cluster sizes, Equation (7) gives smaller rates than Equation (8). Both rates rapidly decrease as relative error

drops less than 1%, when cluster size exceeds $i = 17$, and show a minimum near cluster size $i = 558$–$559$. Figure 1b shows number density curves calculated by the ungrouped CD model, with two emission rates in Figure 1a. Due to having a smaller emission rate, Equation (7) generates a smaller nucleation barrier than Equation (8), which causes larger number densities at small cluster size range. For large cluster size range, the difference is slight because the two emission rates are very close. Consequently, the choice between these two rate equations is not important, especially within a large size range for a long precipitation time. In the following work, emission rates are calculated only using Equation (8), which has better flexibility to add features if necessary.

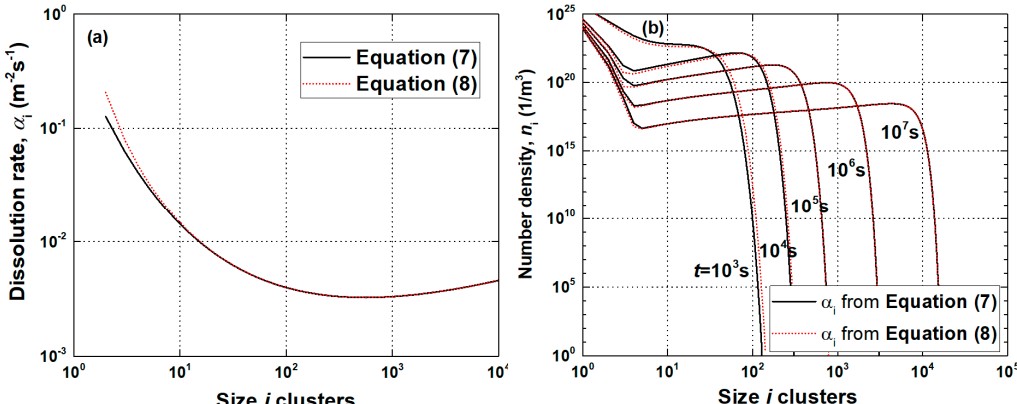

**Figure 1.** Calculation of two dissolution rates and their influence on the evolution of number densities using an exact ungrouped cluster dynamics CD model for $Al_3Sc$ precipitates at 300 °C: (**a**) dissolution rates, (**b**) number density curves.

The calculated number densities of each size group from the ungrouped, and all three grouped, CD models are plotted in Figure 2. For comparison, the number densities of clusters in the ungrouped model are summed into the same size ranges according to Equation (7) by post-processing after the original calculation. These curves are not smooth everywhere, because the size group number range actually varies at some locations. For example, the size group number range changes from $\Delta x_j = 1$ to $\Delta x_{j+1} = 2$ near group $j = 20$, corresponding to a cluster radius $r_j \approx 0.691$ nm. This causes a small jump on all curves near this size. Similar minor glitching behaviors may occur elsewhere when the size group number range changes.

In Figure 2a, the uniform distribution of the Kiritani PSG model tends to overestimate the cluster number densities of a given size group, especially when clusters of this group just start to form. The uniform profile actually overpredicts the number count on whichever side of the size group has the lowest number densities. In this case, the right side towards the tail, is of the most concern for accuracy problems, where precipitate size is largest and most important, and number densities are low. The precipitation is thus forced to propagate more quickly than its actual behavior, which generates a significantly larger precipitate size distribution for this method. A similar conclusion was found by Koiwa when applying this method to CD modeling of quenched-in vacancies; and the distortion of size distribution became even worse with larger size group number ranges [14].

In Figure 2b, the linear distribution of the G-O PSG model generally shows better agreement with the ungrouped CD model, but makes the number densities of some size groups negative. It is important to note that these negative number densities likely come from the assumed linear distribution itself, and cannot be avoided by further decreasing the time step size or providing a more accurate estimation of $J_j^*$. To investigate this, a calculation was performed using a much smaller time step size or the exact number density of each individual cluster in Equation (17), but it showed little improvement of the accuracy, and still exhibited negative number densities, while making the calculation much slower.

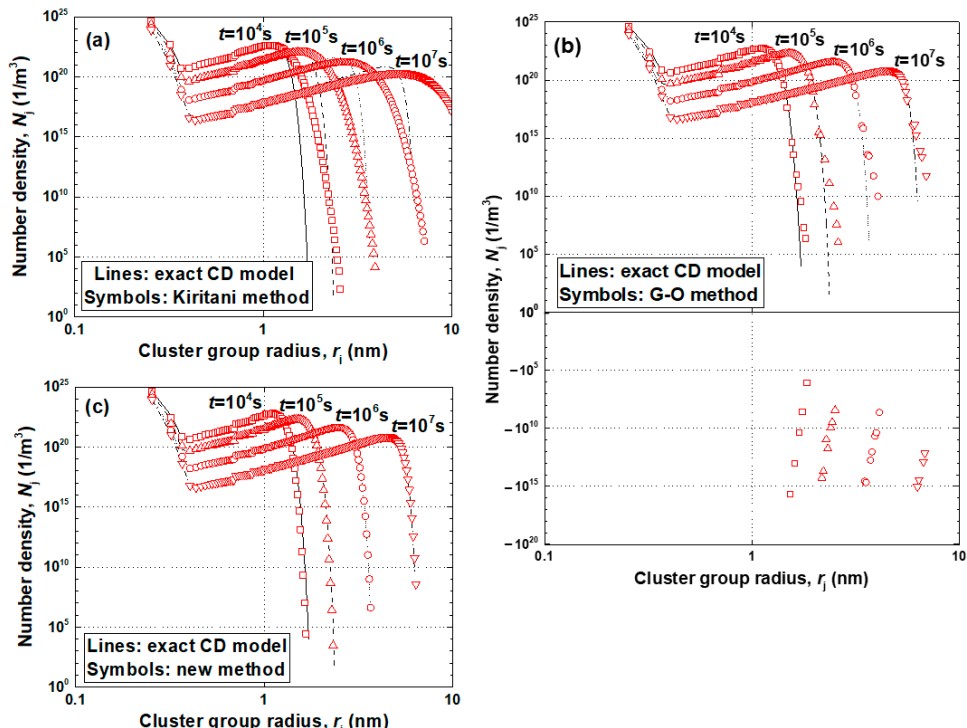

**Figure 2.** Comparison of calculated size distributions of three grouped CD models with the exact ungrouped CD model for $Al_3Sc$ precipitates at 300 °C: (**a**) Kiritani method (uniform distribution), (**b**) G-O method (linear distribution), (**c**) new grouping method (log-linear distribution).

In Figure 2c, the log-linear distribution of the new PSG method suggested in this work is demonstrated to provide very good agreement with the ungrouped CD model. In addition, there are not any negative group number densities.

The computational costs of the PSG methods are shown to have dramatically reduced in Table 1. All the calculations were run on an Intel i7-6700K 4GHz CPU-core (using software: Matlab, 2016, MathWorks, Natick, MA, The United States), 32G RAM, Windows 10 PC to enable a fair comparison. The computational time was smallest for the Kiritani method, with the simplest assumption of uniform cluster number densities within each group. Although the assumed linear distribution of the G-O method is much easier to calculate than the log-linear distribution of the new grouping method, it produces negative group number densities, and only the minimum time step must be used for integration in the later calculation. This obviously affects its efficiency, and causes its calculation time to be even higher than that of the new grouping method.

**Table 1.** Comparison of computational costs of different cluster dynamics models for a precipitation time of $1 \times 10^7$ s at 300 °C.

| Cost\Method | Ungrouped CD Model | Kiritani Method | G-O Method | New Method |
|---|---|---|---|---|
| Computational space | $i_M = 16{,}000$ | $G_M = 200$ | $G_M = 154$ | $G_M = 149$ |
| Computational time | 8472 s | 179 s | 1798 s | 675 s |

Based on the assumed distribution of cluster number densities inside each group, the calculated group number densities of the new PSG method can be transformed to the molecule number densities. Figure 3 compares the molecule number densities of the new grouped CD method, which has the log-linear distribution assumed within the size group, with results for the exact ungrouped CD model. The vertical lines stand for the boundaries of the size groups calculated as $x_j + 1/2$, and the circles on the grouping curves correspond exactly with clusters with an integer number of

molecules. The excellent agreement shown in these results demonstrates that the new log-linear PSG grouping method gives accurate predictions of cluster number densities, even on a molecular scale.

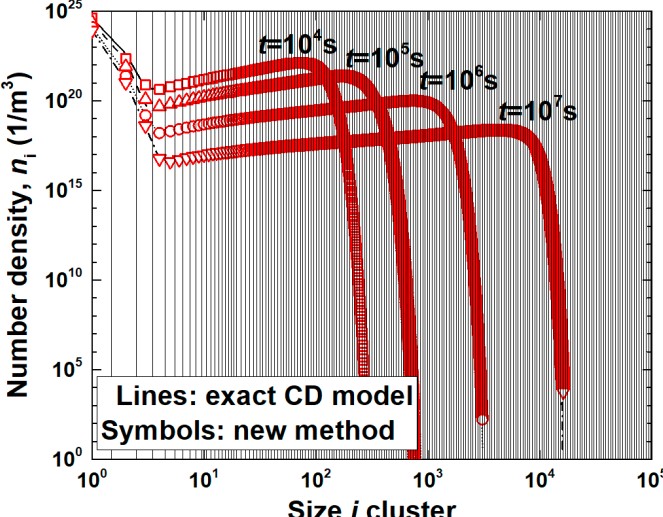

**Figure 3.** Comparison of molecule-based size distributions calculated with the new log-linear grouped CD model, and exact ungrouped CD model for $Al_3Sc$ precipitates at 300 °C.

The total number density and mean radius of precipitated particles calculated with the grouped and ungrouped models are shown in Figure 4. The truncating radius used for matching measurements was set as $r_{tr} = 0.745$ nm (or $i_{tr} = 25$), which approximately agrees with the smallest particle size reported in the TEM images [39]. Since the Kiritani PSG method predicts a larger cluster size distribution, it is not surprising to observe a smaller total number density and a larger mean radius of precipitated particles for this method, which experiences worse accuracy with increasing precipitation time. Both the G-O and new PSG methods gave much better agreement with the exact, ungrouped model solution. This shows that negative cluster number densities do not necessarily ruin the results, which agrees with previous observations [17,18].

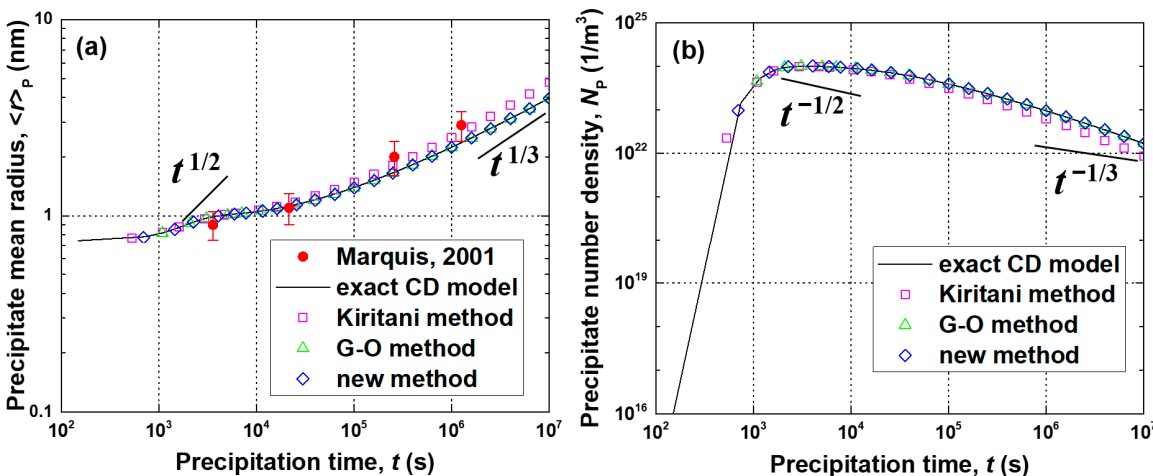

**Figure 4.** Evolution of mean size and total number density of precipitates comparing three grouped CD models with the exact ungrouped CD model for $Al_3Sc$ precipitates at 300 °C: (**a**) mean radius, (**b**) total number density.

At the intermediate growth stage, the slopes of the total number densities roughly match $N_P \propto t^{-1/2}$, but the mean radii exhibit slopes significantly flatter than $\langle r \rangle_P \propto t^{1/2}$, which differs from Zener growth theory [51]. The total number densities and mean radii from all models roughly exhibit the slopes of

$N_P \propto t^{-1/3}$ and $\langle r \rangle_P \propto t^{1/3}$ at the final coarsening stage, which matches Lifshitz–Slyozov–Wagner (LSW) coarsening theory [52,53]. The differences are possibly because all nucleation, growth, and coarsening actually occur simultaneously, and LSW theory is only valid for the very late stages of precipitation, and the strict conditions for establishing both theories are hard to satisfy for practical precipitation.

For the precipitation of Al₃Sc at higher aging temperatures of 350 °C and 400 °C, the ungrouped model adopts the largest number of molecules in a cluster, $i_M = 10^5$, which corresponds to the largest size cluster, $r_{max} = 11.82$ nm, and the computation stops when $n_{i_M-1} \approx 1/m^3$. The evolutions of mean precipitate radius with time calculated by the grouped and ungrouped CD models are compared in Figures 4a and 5 with TEM measurements [39,44]. All four CD models are shown to match reasonably well with the evolution of the mean precipitate radius in most experiments, except for the Kiritani method, which gives a slightly larger size. Relatively poor agreement was found at the highest temperature of 400 °C, which is likely because heterogeneous nucleation is more favored with increasing precipitation temperature [49,50]. In addition, a transition from coherent to semi-coherent precipitation is observed when the radius of Al₃Sc particles exceeds 15–40 nm, and initially spherical precipitates may become cuboidal, or even rod-like [39,42,48–50]. As the CD model in this work is for homogeneous precipitation and ignores these other precipitation mechanisms, a good match with experiments was only expected for small precipitate size and low aging temperature. Although the results of some previous grouping CD models correlated better with experiments than the new PSG model at 400 °C [5–7], they possibly used a similar grouping to the Kiritani method, so the better agreement may be just fortuitous. It is also worth noting that the ungrouped CD model is only effective for a short precipitation time, even with a large molecule number of $i_M = 10^5$. Thus, the grouped models, with their great computational efficiency, are the only option for most practical precipitation simulations.

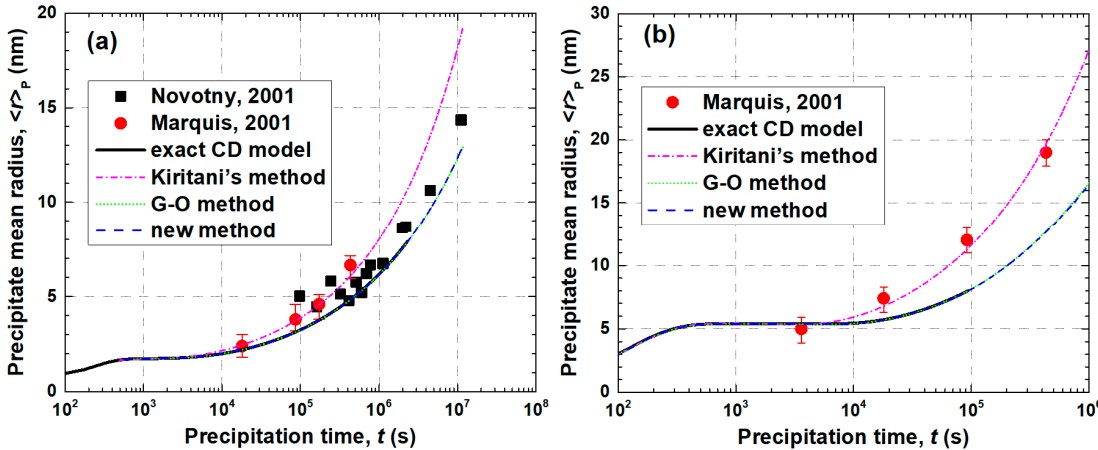

**Figure 5.** Comparison of calculated and measured evolution of mean radius of Al₃Sc precipitates at other annealing temperatures [39,44], (**a**) 350 °C, (**b**) 400 °C.

The calculated precipitate size distributions of the new grouped CD model at different annealing times are compared in Figure 6 with the experimental measurements, as well as with LSW coarsening distributions. The x axis is the normalized particle radius $\bar{r} = r/\langle r \rangle_P$, which is the ratio of particle radius to the mean precipitate radius; and the y axis is the precipitate size distribution function $g(\bar{r}) = (\Delta N/N_P)/(\Delta r/\langle r \rangle_P)$, which is defined as the ratio of the normalized particle number density to the normalized particle size interval. By these definitions, experimental or numerical distributions with different size intervals can be readily compared, so long as the mean precipitate sizes have been validated separately (which was done in Figure 5). It is concluded that the distributions obtained from the new PSG CD model agree well with the measured size distributions [44]. In addition, their asymptotic limit matches the LSW distribution. Finally, the accuracy improves with increasing precipitation time.

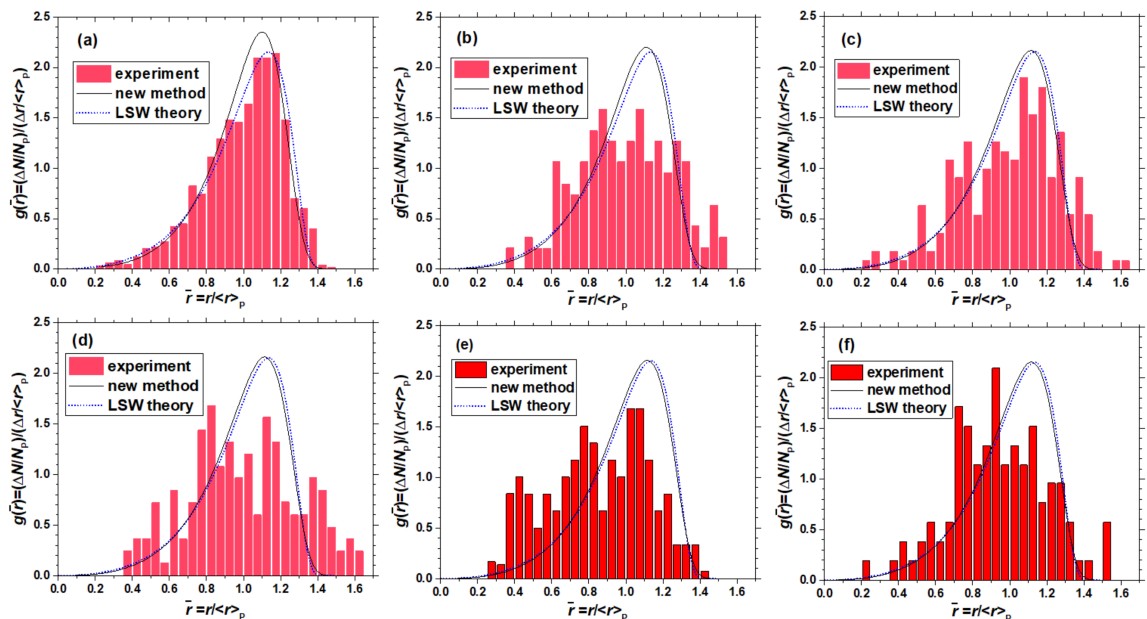

**Figure 6.** Comparison of normalized precipitate size distributions of Al$_3$Sc precipitates at 350 °C from new grouped CD simulation with experimental measurements [47], and LSW distribution (**a**) 27 h, (**b**) 192 h, (**c**) 552 h, (**d**) 624 h, (**e**) 1251 h, and (**f**) 3093 h.

## 6. Conclusions

(1) A new cluster dynamics model is presented for predicting precipitation formation, which is shown capable of reproducing many important metallurgical phenomena, such as matching the Gibbs–Thompson equation.

(2) Three different size grouping methods, including the new method, were evaluated by comparing with the exact solution from the ungrouped CD model for Al$_3$Sc precipitation in an Al-0.18 at.% Sc alloy. Assuming a uniform distribution of cluster number densities within each size group tends to generate a broader size range and a larger mean precipitate size. Assuming a linear distribution gives better agreement with the total properties of precipitates, but may produce negative number densities for some size groups, so needs special treatment. The new PSG model, assuming a log-linear distribution within each size group, produced the best match with both total and local properties, compared with the exact solution.

(3) The mean precipitate size calculated with the new PSG CD model matched reasonably well with the experimental measurements for different aging temperatures. In addition, the new model accurately predicted the normalized size distributions, which agreed with the experiments, and correlated better with LSW coarsening distribution with increasing aging times.

(4) The new PSG CD model is more accurate than other methods for a given discretization and time-integration method, which enables it to be more computationally efficient, and promising for dynamic modeling of realistic large-scale precipitation phenomena.

(5) (5) Finally, future work is still needed to make this methodology more useful, such as considering the transition from coherent to semi-coherent precipitation, heterogeneous precipitation on dislocations and grain boundaries, and multiple alloy phases.

**Author Contributions:** Conceptualization, methodology, writing—original draft, K.X.; writing—review & editing, B.G.T.; software, validation, Y.W. and H.K.; project administration, visualization, H.W. and Z.W. All authors have read and agreed to the published version of the manuscript.

**Funding:** This research was funded by the National Natural Science Foundation of China (Grant No. U1960109, U1760117, 51974004) and Key Program for the Innovation and Entrepreneurship Support Plan for Returning Overseas Chinese Scholars in Anhui Province.

**Conflicts of Interest:** The authors declare no conflict of interest.

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
