# Peer review of "Grouping Methods of Cluster Dynamics Model for Precipitation Kinetics"

_metals, doi:10.3390/met10121685_

Round 1

Reviewer 1 Report

In the manuscript, the authors suggested a particle size grouping method of cluster dynamics with logarithmically-linear distribution of cluster number densities within groups. They compared results of the suggested method with results of other grouped CD models and ungrouped CD model for Al3Sc precipitation in Al-Sc alloy.

The paper under review is interesting and after several minor revisions it can be recommended for publication.

  1. t from equation (1) is not defined
  2. 2. line 110: Authors mentioned equation (14) from Clouet’s work [6]. It is not appropriate to refer to an equation from another work in this way. If necessary, an equation should be stated.
  3. line 193: Authors write that Equation (22) corrects minor errors. What are the errors?
  4. In the manuscript, # character is used. (for example on lines 283, 288…) Please, correct it.
  5. On line 415, TEM images is mentioned. However, a picture or reference to the literature is missing.
  6. line 437: incorrect figures numbers
  7. How do you explain that Kiritani method correlates best with experimental results according to Novotny in Fig 5? 
  8. conclusion 3 and Fig. 6: The PSG method correlates better with the LSW distribution with increasing time. However, it does not improve over time compared to the experiment. (Fig. 6)
  9. I recommend enlarging the figures.

Reviewer 2 Report

In this paper the authors propose a some variant of the cluster dynamic (CD) method using a particle-size-grouping (PSG) approach  to study  growth and coarsening kinetics in alloys. After introducing the main assumptions of the CD method,  it is shown that the PSG approach coupled with the CD method open a way to model the later stages of  precipitation kinetics where a large distribution of clusters size can reduce the relevance of the “classical” CD method. In particular, the three ways to construct the distributions of cluster number densities are considered: uniform, linear and a new logarithmically-linear distribution. Then, this new type of distribution  is tested to describe the main characteristics of the precipitation kinetics in Al-Sc system.  

   In general, the paper is good in concept. This is an interesting kernel of work, but some arguments need to be polished before the results are convincing.  I think this article can be considered for publication in Metals once the authors are tightened up the model description and the discussions.

-         In the introduction the authors claim that the CD method is more appropriated method to describe the precipitation kinetics. I agree that to describe the clustering, especially in irradiated materials, or  first stages of precipitation the CD Method can be very useful approach. However, I am not  agree that this method is a good choice to model coarsening kinetics. I think that at present the phase field approach is maturing to become a universal tool for modelling precipitation kinetics in material science, especially its coarsening stage.  The authors should clearly explain what is the advantage to use the CD approach to model a microstructural evolution.

-       It is not clear  how the choice of J* impacts  the final results (cluster size distribution, mean radius of precipitate,…). Globally, it will be important to show  how the parameters of model ( λ, ζ, number of groups, number of atoms in each group) influence the simulation results .

-       Lines 279-281   “In this work, the maximum size of molecules or groups was fixed to a constant large number through the entire calculation, but the evolution of a given cluster or group is calculated only when its cluster number density becomes larger than the minimum given in Eq.(36)”         For that sentence  it’s not clear for me how the authors fixed the number of atoms in the cluster and the number of groups. This point should be clarified.

-       I have  doubts about the pertinence  of proposed model to simulate the size distribution of L12 precipitates in Al-Sc alloys at coarsening stage. Despite a low misfit between the lattice parameters of Al3Sc ordered particles and disordered matrix the approximation of spherical particle at later stages of kinetics doesn’t work. In general case, the contribution of elastic energy increases with increasing of the volume of precipitate and strongly influences a coarsening kinetics.  As follow from experiment (for example, D. Seidman Acta Mat. 49, 2001) in Al-Sc system during particle growth the initially spherical precipitates become cuboidal and sometimes have more complex, like dendrite, structure. This shape instability is due to the elastic energy and plays an important role in size distribution of precipitates. As example, this is a main reason why the size distribution in this system is so large at later stages of coarsening. So, from my opinion the CD method becomes non appropriated approach to describe this kind of dynamics.

-       In simulation , the initial configuration contains some number of clusters with a critical radius. In this case how the real time of simulation was calculated and compared with experimental results ? How was estimated the nucleation time ?

-       The regions where the mean radius grows as t1/2 or t1/3 is not well defined in Fig.4

-       The  details concerning the experimental part of this work should be better defined.
